# An evaluation to determine if reading the mind in the eyes scores can be improved through training

**Jacklin Hope Stonewall**[1☯*], **Kaitlyn M. Ouverson**[2☯], **Andrina Helgerson**[1¤‡], **Stephen B. Gilbert**[1‡], **Michael C. Dorneich**[1‡]

**1** Department of Industrial and Manufacturing Systems Engineering, Iowa State University, Ames, Iowa, United States of America, **2** Department of Psychology, Iowa State University, Ames, Iowa, United States of America

☯ These authors contributed equally to this work.
¤ Current address: Emerson, St. Louis, Missouri, United States of America
‡ AH, SBG and MCD also contributed equally to this work.
* jacklins@iastate.edu

**Data Availability Statement:** All data files are available from the Open Science Framework database (https://osf.io/bd4rp/).

## Abstract

The Reading the Mind in the Eyes Test (RMET) has received attention due to its correlation with collective intelligence. If the RMET is a marker of collective intelligence, training to improve RMET could result in better teamwork, whether for human-human or human-AI (artificial intelligence) in composition. While training on related skills has proven effective in the literature, RMET training has not been studied. This research evaluates the development of RMET training, testing the impact of two training conditions (Naturalistic Training and Repeated RMET Practice) compared to a control. There were no significant differences in RMET scores due to training, but speed of response was positively correlated to RMET score for high-scoring participants. Both management professionals and AI creators looking to cultivate team skill through the application of the RMET may need to reconsider their tool selection.

## Introduction

Imagine you find some delicious cookies in the kitchen pantry. You hide the cookies under the sink so your sibling does not eat them. When your sibling enters the kitchen, where will they look for the cookies? If you say in the pantry, you have demonstrated Theory of Mind (ToM). In other words, ToM allows one to think through the example from the sibling's perspective to determine where they would look for the sweets. Theory of Mind is defined as the ability to attribute mental states to ourselves and others [1, 2]. Humans use this unique ability to predict or understand another's behavior [3]. Young children typically have not yet developed ToM and will expect a sibling to look under the sink, as they assume the sibling holds the same knowledge they do within their minds [4, 5]. The term "ToM" is sometimes referred to as "Mentalizing," or (the more magical-sounding) "mindreading" [6]; however, the authors in the present paper will use 'ToM' as an umbrella term referring to all three, simultaneously.

**Funding:** The author(s) received no specific funding for this work.

Many social and interpersonal skills associated with ToM have been linked to team success [7]. As more classrooms and workplaces acknowledge the benefits of better learning outcomes through group and team-work, and as enterprises continue to pursue human-AI (artificial intelligence) teaming, efforts to understand factors affecting these collaborations have also gained momentum. Research shows that increased ability to understand another's feelings or perceptions aids in the ability to manage social situations that are critical to successful team-work [8, 9]. In human-AI interactions the machine's ability to attribute mental states to others greatly enhances the quality of the interaction [10]. As such, AI researchers have turned to ToM as a way to provide automation with the ability to process human facial expression data in real time [10]. Likewise, the ability of humans to utilize ToM to manage human-AI interactions also needs to be studied. This requires the capability to measure ToM and to understand if a ToM-based ability to succeed in cooperative tasks can be increased through training.

ToM may be measured in various ways, including the Reading the Mind in the Eyes Test (RMET). This test of social sensitivity evaluates individuals' ability to "tune in" to another's mental state by looking at images of the eye region of the face and matching the expression they see to the closest-matching one-word description [11]. In this way, participants demonstrate their ability to attribute a mental state to others.

Scores on the RMET correlate positively with collective intelligence, or an individual's ability to succeed in a variety of cooperative tasks [9, 12]. However, it is unknown whether an improvement on RMET scores leads to an improvement in collective intelligence. The first step in investigating if this is the case is to establish whether or not it is possible to improve RMET scores through training. While training on ToM is commonly used in the treatment of autism [13], training designed specifically to increase individual ability to glean mental state information from the faces of others has not been developed. By developing better "face-reading" skills, it is hoped that individuals (human or otherwise) could thus be trained to be better teammates. Therefore, the aim of this work was to develop and test methods of training individuals to perform better on the RMET.

The RMET is a widely applied and accepted test for measuring ToM [14, 15]. The RMET presents participants with 36 photographs of the eye regions of given individuals. Paired with each photograph are four descriptor words from which the participant must select the one best corresponding with the emotion exhibited in the eyes. Participants score one point for every correct answer, with a maximum score of 36. Evaluations against other tests for measuring ToM have shown that RMET has fair reliability [14, 15].

There are a few known limitations of the RMET: neurodiversity, cultural differences in emotion recognition and processing, and knowledge of the English language. First, and perhaps most clearly, the RMET has demonstrated a strong ability to identify individuals with different social intelligence but otherwise typical cognitive intelligence; examples include neurodiverse individuals, such as those with high functioning autism or Asperger syndrome [11, 14]. This was the purpose for which the RMET was originally developed [11].

While the basic emotions (happiness, sadness, surprise, fear, disgust, anger, and contempt) are accepted as universal by some [16–18], research by Jack et al. [19] countered this belief by demonstrating a cultural influence on how individuals express different emotions. Jack and colleagues' [19] research found that for East Asian individuals, whose facial expressions are governed by different, culture-specific rules for how emotions should be displayed, Ekman's basic emotions were not consistently identified, and other emotions fundamental to the culture were not included in these "basic emotions." Further, the RMET features exclusively light-skinned faces which could affect the ability of non-white participants to recognize the

emotion displayed in the stimuli, reminiscent of the "other-race effect," in which individuals have difficulty recognizing faces of individuals whose race differs from their own [20].

Lastly, The RMET involves the association of facial expressions to emotion words which may be difficult for participants whose native language is not English. The difficulty of completing the RMET outside of one's native language and the utility of offering the test in multiple languages is evidenced by the translation of the test into French [21] and Spanish among others [22].

Various studies have been conducted on increasing ToM through training, including Kidd and Castano [23] in which a correlation was identified between reading literary fiction and improved RMET scores. Participants in Kidd and Castano's [23] experiments read one brief text and showed only short-term improvements to RMET scores, a result that is debated in the literature [24–26]. Studies which have looked into establishing long-term improvements have focused primarily on children with more-extensive (compared to Kidd and Castano's work [23]) literature-based training [27, 28]. These studies found that by discussing stories filled with mental-state vocabulary, children's ability to understand this vocabulary and to interpret the emotions of others was improved. However, limited studies have been performed on ToM training for adults [29, 30], the purpose which the RMET was created to fulfill [6]. Further, understanding how RMET may or may not train ToM may help to establish deep-learning strategies for teaching AI to infer mental states [10].

The study had four main hypotheses. ToM training has shown to be helpful for those with diminished ToM capacity. As RMET is a Theory of Mind measure [11], a Naturalistic RMET Training was developed based on traditional ToM training. By basing Naturalistic RMET Training on ToM training, it is hypothesized that individuals who receive training will improve their RMET scores.

*H1*: *Naturalistic RMET Training will result in a RMET score increase compared to No Training.*

To ensure that the effect of the Naturalistic Training is more than any changes in RMET score due to familiarity with the content of the test, a portion of participants were placed in the Repeated RMET Training group, where they were instructed to take the test multiple times without feedback. The Naturalistic Training is hypothesized to result in a larger score increase by teaching participants to read emotions, rather than just expecting them to correct their ability without feedback.

*H2*: *Naturalistic RMET training will result in a greater increase in score than Repeated RMET training.*

Participants with higher initial RME ability are hypothesized to benefit less from training than those whose initial RMET ability was low.

*H3*: *Low initial RMET performers will see greater improvements in RMET score from training than those who initially scored high on the RMET.*

We expect that RMET score will be higher for individuals who answer the questions faster, i.e., in less time. This hypothesis was first published by Tracy and Robins [31] as part of their second study in that manuscript. The present study does not restrict response time or induce cognitive load, instead allowing participants to deliberate and observing whether, within an environment encouraging of deliberation, higher average response times per question were related to lower RMET score.

*H4*: *Time per question is negatively correlated to RMET score.*

## Experimental method

The objective of the study was to determine if training can impact RMET scores. This study was approved as exempt by the Institutional Review Board of Iowa State University (#18–075). Electronic informed consent was obtained from all participants.

### Participants

The study included 429 participants (307 women, 117 men, 3 non-binary) recruited from a public university and social media. Participants averaged 30.4 years of age (range 18–74). English was the most comfortable language for 76% of participants and 73% of participants identified their native country as the United States. In terms of sexual orientation, 90.4% of participants identified as heterosexual or straight, 0.7% identified as lesbian, 1.2% identified as gay, 3.5% identified as bisexual, 2.3% identified as an orientation not listed, and 1.9% chose not to respond. For completing the study, participants were given a chance to win one of three $99 Amazon e-gift certificates.

### Experimental design and procedure

The experiment was a between-subjects design in the form of an online survey via Qualtrics. All participants were instructed to complete the experiment on a laptop or desktop computer and the use of smartphones or tablets was discouraged. Participation in the experiment lasted approximately 45 minutes. The specifics of the procedure are discussed below.

Once participants had given electronic consent and verified their age, they completed the Pre-training RMET (henceforth referred to as the Pre-RMET). After completing the Pre-RMET, each participant was randomly assigned to one of three conditions, Naturalistic Training, Repeated Training, or No Training, using Qualtrics's randomization function. Time in the assigned condition was approximately 20 minutes for all conditions. In the "Repeated Training" condition, participants repeated the RMET three times without feedback. In the "Naturalistic Training" condition, participants were guided through the training described below, which was designed to improve their ability to read faces. In the "No Training" condition, participants were given unrelated distractor tasks in the form of visual puzzles and videos to keep them occupied for the same amount of time taken in other training groups. The distractor tasks (I-spy puzzle, spot-the-difference puzzle, videos to watch) were specifically chosen to be as unrelated to the test as possible yet require visual processing similar to the RMET. After each distractor video, the participant completed a quiz on the content to allow for control for participant inattention. All participants then completed the Post-training RMET (Post-RMET) and a demographics survey.

Participants in the Naturalistic Training condition were introduced to the concept of basic emotions by watching a video on decoding facial expressions [32] and reading an article which explains those facial expressions in more detail [33]. Drawing on the Theory of Mind training described by Adibsereshki, Nesayan, Asadi Gandomani, & Karimlou [34], in which participants were given feedback on how well they sorted pictures and drawings of facial expressions into emotion categories, all of the Naturalistic Training media were followed by quizzes with correct/incorrect feedback.

Training on complex emotions featured two Pixar short films, *Lifted* [35] and *Partly Cloudy* [36]. After watching these short films, participants answered questions about the animated characters' emotions based on still images from the films. Naturalistic Training ended with a sample RMET-like quiz that used dynamic images, or video clips, of an eye-region of faces expressing emotions as stimuli, rather than static images [37], and feedback was given on the answers to each question. The next section briefly describes the systematic methods used to

develop these training stimuli, while more detail can be found in Ouverson, Stonewall, Gilbert and Dorneich [38].

### Naturalistic training stimuli development

The objective of the Naturalistic Training (previously referred to as Strategic Training; [38]) was to give participants practice interpreting facial cues, patterns, and expressions so that they could ultimately interpret the emotional state of the person. To develop the materials, a set of training stimuli different from the 36 faces in the RMET was needed. As detailed by Ouverson et al. [38], 92 complex emotion answer choices were assigned from the original RME to the Pixar stills. First, the answer choices from the RMET were grouped into seven categories corresponding to basic emotions: Anger, Sadness, Surprise, Happiness, Disgust, Fear, and Contempt. Each category had approximately 20 complex emotions answer choices listed (some words appeared in multiple categories). Stills were also uniquely sorted into the seven basic emotion categories. A survey randomly assigned a Pixar still to four of the 20 RMET answer choices corresponding to the basic emotion category. A separate sample of participants from this paper's primary study ($n$ = 136) rated how well the choices fit with the emotion seen (1 = "Does Not Fit at All" and 7 = "Fits Very Well"). The answer choice with the highest mean score was chosen as the "correct" answer for the still, while the three answer choices with the lowest mean scores were chosen as the distracter choices. The results from this initial survey were used to create the Naturalistic Training on reading complex emotions.

### Independent variable

The study had one independent variable, *Training Type*, with three levels: Repeated Training (the RMET was taken three times without feedback), Naturalistic Training (participants were trained using different materials with feedback), and No Training (participants completed distractor tasks in lieu of training).

### Quasi-independent variable

A quasi-independent variable was *Initial RME Ability* (low, high). Initial RME ability was calculated using the Pre-RMET score. If a participant scored at or below the 25th percentile of collected data (i.e., the observed first quartile, a score of 25 or less), they were considered to have low RME ability. Conversely, if the participant achieved a score of at least 30 (75th percentile), they were considered to have high RME ability. In order to observe the effects of high and low, but not moderate Initial RME ability, the middle quartiles were not included in analysis.

### Dependent variables

The study included three dependent variables: Post-RMET score (out of 36), Change in RMET score (range of -36 to +36), and Time per Question (in seconds; measured for each RMET question). The Post-RMET was replicated in Qualtrics after training using the same images and choices as the pre-training RMET and scored out of 36. Change in RMET score was calculated during analysis by subtracting the initial RMET score from the final RMET score and used to delineate "low-scoring" and "high-scoring" participants. Because participants are introduced to manipulations between the Pre- and Post-RMET, the RMET scores compared in most analyses are the Pre-RMET scores.

## Data analysis plan

After assumptions of normality and homogeneity of variance were met, ANOVAs and Tukey's Honest Difference (HSD) Post-hoc tests were used to analyze the data. Results at the $p < .05$ level are reported as statistically significant, while marginal significance is assigned to those at the $p < .10$ [39]. Kendall's tau-b was calculated to assess the relationship between Time per Question (a continuous variable) and RMET score (an ordinal variable) for both the Pre- and Post-RMETs. Cohen's $d$ was used for effect size. Cohen [40] indicated that, when interpreting effect sizes, $0.2 \leq d < 0.5$ showed a small effect, $0.5 \leq d < 0.8$ could be considered a medium-sized effect, and $d \geq 0.8$ showed a large effect, an interpretation the authors adopt in the present work.

## Data integrity

To ensure maximum data integrity, the researchers removed data that failed to meet a set of criteria designed to ensure that the remaining participants were ones who were assumed to have treated the survey seriously:

1. Participants must answer at least 75% of questions on each scale.

2. Participants must spend at least 940 seconds (just under 16 minutes) on the entire survey.

3. Participants must stay on the Naturalistic Training videos at least long enough to see all important content (skipping the video credits will not result in removal).

4. Participants must score at least six of eight points on the distractor video quizzes (random guessing resulted in a score of four, so this higher threshold ensured attention).

5. Participants must take at least half the time that the researcher in charge of choosing the puzzles took to complete those distractor activities (considered the minimum time).

After 100 participants were excluded via the criteria above, the Naturalistic Training condition contained 118 participants, the Repeated Training condition contained 145 participants, and the No Training/Control condition contained 166 participants. Demographics by condition are presented in Table 1. The randomization resulted in groups that have roughly similar representation across demographic groups in each condition.

# Results

## RMET training

Fig 1 illustrates the pre- and post-RMET score by condition. The mean Pre-RMET score across all conditions was 27.6 ($SD = 3.5$) while the mean Post-RMET score across all conditions was 27.8 ($SD = 4.2$). There were no significant differences between mean Pre-RMET score and mean Post-RMET score for any of the training conditions.

## Change in RMET performance for high-scorers vs. low-scorers

A two-way ANOVA was run to assess the change in RMET performance for individuals who scored in the top quartile on the initial RMET ($M = 31.7$, SD = 1.3) and those who scored in the bottom quartile on the initial RMET ($M = 23.0$, SD = 2.0). The test revealed a statistically significant difference in change in scores between high ($M = -1.45$, SD = 2.9) and low ($M = 1.7$, SD = 4.1) initial RMET ability ($F(1, 232) = 48.38$, $p < .001$, $d = 0.17$) and a marginally significant difference between training conditions ($F(2, 232) = 2.74$, $p = .07$). A post-hoc Tukey HSD test showed that the Naturalistic Training group's change in RME score was marginally

**Table 1. Demographics of participants per condition.**

| Training Group | Gender (count) | Median Age (IQR) | Native English Speakers (count) | Race/Ethnicity (count) | US-Natives (count) |
|---|---|---|---|---|---|
| Control | F(118) M (48) | 22 (14) | Native(155), Non-Native(11) | White(140), Asian(10), Hispanic/Latino(7), Black/African American(5), Indigenous American/Native Hawaiian(1), Other(3) | Native(149), Non-native(17) |
| Repeated RMET | F(107) M (38) | 23 (15.8) | Native(135), Non-Native(10) | White(118), Asian(5), Hispanic/Latino(10), Black/African American(6), Indigenous American/Native Hawaiian(2), Other(4) | Native(135), Non-native(10) |
| Naturalistic | F(83) M(35) | 28 (22) | Native(108), Non-Native(10) | White(98), Asian(8), Hispanic/Latino(4), Black/African American(4), Indigenous American/Native Hawaiian(0), Other(4) | Native(99), Non-native(19) |

significantly greater than the Control group's change ($p$ = .07, 95% CI (-2.60, 0.07)). The interaction term was not significant. Fig 2 visualizes these differences.

Two ANOVAs were used to inspect the marginal effects of training condition at each level of Initial RME ability, with significance evaluated at an alpha level of .025 to adjust for the use of multiple tests. For those participants who presented with Low Initial RME ability ($F(1, 108)$ = 2.418, $p$ = .12; $M$ = 1.7, SD = 3.4) and those who scored initially high (High Initial RME ability) ($F(1, 128)$ = 0.59, $p$ = .59; $M$ = -1.45, SD = 2.9), the change in RMET scores was not statistically significant.

## Time per question by RMET score

A Kendall's tau-b correlation was used to evaluate the relationship between RMET Score and average time per RMET question, in seconds ($Mdn$ = 9.5; $IQR$ = 3.5–15.5). While RMET Score and time per question were not significantly associated in the pretest ($\tau_b$ = -0.001, $p$ = .97), there was a marginally significant, weak, and negative association in the post-test ($\tau_b$ = -.06, $p$

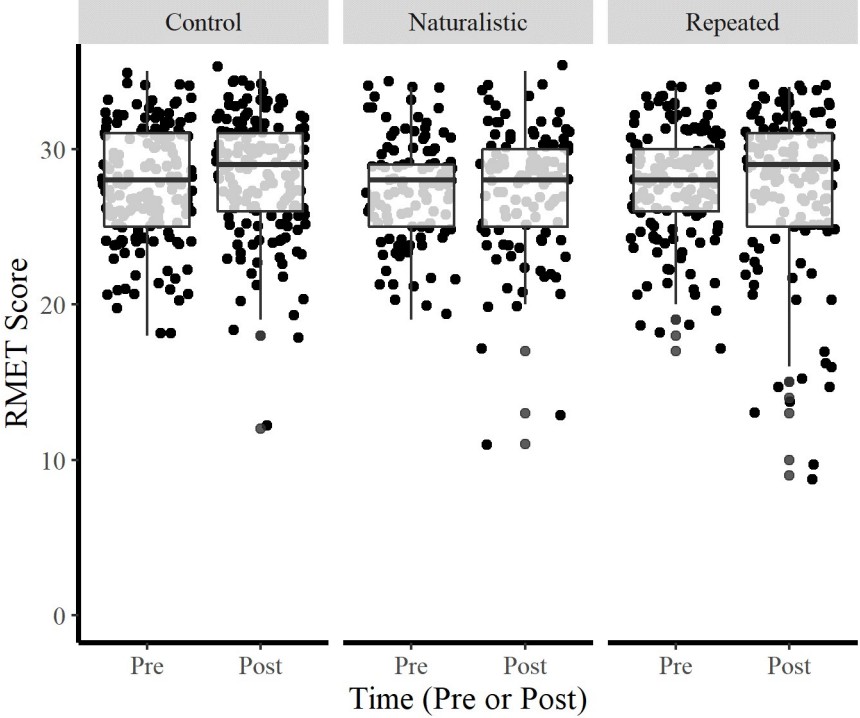

**Fig 1. Pre- and Post-RMET score by condition.** Bars represent one standard deviation from the mean.

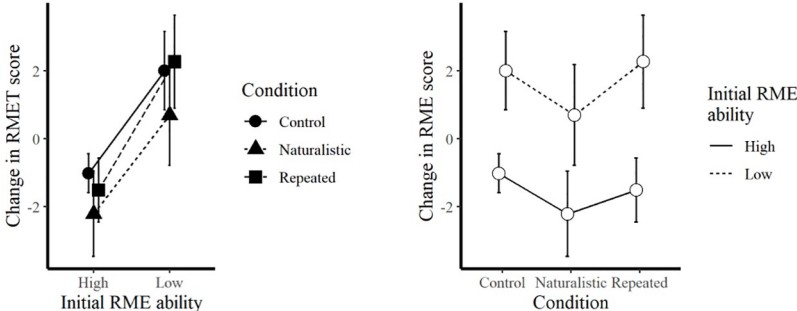

**Fig 2. Difference in mean change in RMET score by condition and initial RME ability.** Error bars represent 95% Confidence Intervals.

= .07). Examining this association for both low- and high-skill RME participants separately, no association was found between overall RMET Score and overall average time per question for low-skill participants ($\tau_b$ = 0.002, $p$ = .98), but a marginally significant, weak, and negative association was uncovered between overall RMET Score and overall average time per question ($\tau_b$ = -.11, $p$ = .09) for high-skill participants.

## Discussion

Hypotheses H1 and H2 were not supported. There were no significant changes between Pre- and Post-RMET for any of the three training conditions. The lack of effect, specifically for the Naturalistic Training condition, could imply that it is difficult to train individuals to perform better on the RMET. This could imply that the skills needed to succeed at the RMET are linked to those social skills.

The Naturalistic Training used in this study was based upon standard ToM training [34]. When used to help autistic individuals, ToM training is administered in person and by a professional. The method of delivery used in this experiment could have therefore impaired the effectiveness of the training. More work is necessary to decide conclusively whether any type of training can improve an individual's ability to "read faces".

Similarly, the ToM training that informed the Naturalistic Training is generally only administered to children. While some studies have indicated that standard ToM training can be effective for older adults [29, 30], little work has examined whether ToM training can be effective for younger to middle-aged adults. Given the average age of the participants in the Naturalistic Training group ($M$ = 32.6 years) and the basis for the Naturalistic Training, the training may not have been effective for the population studied.

Hypothesis H3 was not supported. While those with low initial RME ability saw changes which approached an acceptable level of significance, the result was not statistically significant. This could indicate that the training is simply not effective for people who are already good at Reading the Mind in the Eyes, or it could be evidence of regression to the mean, a statistical phenomenon showing that high performers regress toward the mean or average while low performers will more typically see performance gains relative to their starting measurement [41]. Regression to the mean, in this context, speaks to the observation that average scores for the top 25% performers were on average one-and-a-half points lower (1.45) in the post-RMET, while the bottom 25% saw net gains of near two points (1.7), bringing each group closer to the average than where they started.

Hypothesis H4 was partially supported. All explored associations between average time per question and RMET score were negative, except for the low-skill individuals. While no results

reached true statistical significance, the marginally significant results suggest a need for further testing. The data possibly suggest that emotion-state reading is a function of "fast-thinking." In addition, the lack of a result for the low-skill participants, when comparing average response speed and RMET score, suggests that low-skill individuals may not have the proper schemata to classify the emotions in a fast manner. Therefore, no association between speed and score would be expected.

The researchers note a few limitations of the current research. First, because the survey was distributed over the Internet, the experiment was not as tightly controlled as one distributed in a lab setting. Additionally, participants in Naturalistic Training could pause the videos. In sum, despite the researchers' best efforts, completion times and medium of interaction may have varied from participant-to-participant.

Second, the RME describes specific emotion states using words that were potentially problematic for non-native English speakers. The nuanced differences between, say, "despondent" and "dispirited" led to challenges during Naturalistic Training development.

Lastly, it could be that the sample may have contained more naturally good mind-in-the-eyes readers than the general population. Cursory analysis revealed a significant difference between RMET performance for this study's sample (Mean = 27.6, $SD$ = 3.5) compared to the general population (as reported by Baron-Cohen et al. [11], Mean = 26.2, $SD$ = 3.6, $t(132.4)$ = 2.82, $p$ = .01, but not compared to their student population, Mean = 28.0, $SD$ = 3.5, $t(113.4)$ = -0.75, $p$ = .30.

## Conclusion

The aim of this work was to develop and test methods of training individuals to perform better on the RMET. The design of the training was informed by standard Theory of Mind (ToM) training often used as a way to help autistic people address some of the more difficult components of their neurological difference. As ToM has been recognized as a pivotal component of creating more human-friendly AI, RMET and other methods of training ToM seem like logical sources of training data or strategies. The results of the study indicated that there was no effect of this training on RMET score, but the training's effectiveness may have been impacted by its delivery mechanism. Overall, the training was not shown to be effective. Before using the RMET for evaluation or training, generalization issues must be addressed. Further, eye-region imagery alone may not be enough to teach ToM, whether to human- or AI-agents. In future work related to training development, the delivery of training could be in-person to better echo proven ToM training methods and may need to occur over a period of days to weeks, and especially in the case of training machine ToM, ethical considerations around the use of imagery of white faces, cultural differences in emotions, and other concerns must be addressed.

## Author Contributions

**Conceptualization:** Kaitlyn M. Ouverson, Stephen B. Gilbert, Michael C. Dorneich.

**Data curation:** Jacklin Hope Stonewall, Kaitlyn M. Ouverson.

**Formal analysis:** Jacklin Hope Stonewall, Kaitlyn M. Ouverson.

**Investigation:** Jacklin Hope Stonewall, Kaitlyn M. Ouverson, Andrina Helgerson.

**Methodology:** Jacklin Hope Stonewall, Kaitlyn M. Ouverson, Stephen B. Gilbert, Michael C. Dorneich.

**Project administration:** Jacklin Hope Stonewall, Kaitlyn M. Ouverson, Stephen B. Gilbert, Michael C. Dorneich.

**Supervision:** Jacklin Hope Stonewall, Kaitlyn M. Ouverson, Stephen B. Gilbert, Michael C. Dorneich.

**Writing – original draft:** Jacklin Hope Stonewall, Kaitlyn M. Ouverson, Andrina Helgerson.

**Writing – review & editing:** Jacklin Hope Stonewall, Kaitlyn M. Ouverson, Andrina Helgerson, Stephen B. Gilbert, Michael C. Dorneich.

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
