## [Decision Letter · Decision Letter 0]

6 Oct 2021

PONE-D-21-26174An Evaluation to Determine if Reading the Mind in the Eyes Scores Can Be Improved Through TrainingPLOS ONE

Dear Dr. Stonewall,

Thank you for submitting your manuscript to PLOS ONE. After careful consideration, we feel that it has merit but does not fully meet PLOS ONE’s publication criteria as it currently stands. Therefore, we invite you to submit a revised version of the manuscript that addresses the points raised during the review process. Both Reviewers are quite critical with regard to the way the manuscript is written and I feel it would need some substantial restructuring before it is ready for publication. However, I also think there is scope for revision and therefore, would like to offer you the opportunity to resubmit the revised paper, if you feel you can address the points both Reviewers make.  Please submit your revised manuscript by Nov 20 2021 11:59PM. If you will need more time than this to complete your revisions, please reply to this message or contact the journal office at plosone@plos.org. Please include the following items when submitting your revised manuscript:A rebuttal letter that responds to each point raised by the academic editor and reviewer(s). You should upload this letter as a separate file labeled 'Response to Reviewers'.A marked-up copy of your manuscript that highlights changes made to the original version. You should upload this as a separate file labeled 'Revised Manuscript with Track Changes'.An unmarked version of your revised paper without tracked changes. You should upload this as a separate file labeled 'Manuscript'.

We look forward to receiving your revised manuscript.

Kind regards,

Magdalena Ewa Król, Ph.D.

Academic Editor

PLOS ONE

Journal Requirements:

4. Please ensure that you refer to Figure 1 and 3 in your text as, if accepted, production will need this reference to link the reader to the figure.

Reviewers' comments:

Reviewer's Responses to Questions

**Comments to the Author**

1. Is the manuscript technically sound, and do the data support the conclusions?

Reviewer #1: No

Reviewer #2: Yes

2. Has the statistical analysis been performed appropriately and rigorously? 

Reviewer #1: Yes

Reviewer #2: Yes

3. Have the authors made all data underlying the findings in their manuscript fully available?

Reviewer #1: No

Reviewer #2: Yes

4. Is the manuscript presented in an intelligible fashion and written in standard English?

Reviewer #1: Yes

Reviewer #2: No

5. Review Comments to the Author

Reviewer #1: General comment

In an online study, the authors tested how training “face based mindreading” in participants could help them performing better at the Reading the Mind in the Eyes Test (RMET). They included two control conditions: “repeated training” (performing the RMET 3 times in a row) and “distraction” (visuospatial tasks unrelated to mindreading). They also explored how sociodemographical variables such as self-declared gender, race, height, etc. influence the results, with the underlying hypothesis that marginalized groups (females, LGBTQ+ people, POC, etc.) could be better motivated to mindread faces in their environment. They found no effect of training. Some demographical variables showed the expected effect (gender, race, native language), but not others.

I really appreciated the good English of the manuscript, and how the authors explored variations of cognitive skills based on social and cultural status.

But I found the manuscript to be very confusing from a theoretical point of view. A lot of problematic assertions are backed up with dated literature, or sometimes are not even justified or referenced. There are some mistakes on how hypotheses and results are articulated. Also, it seems that there is no cohesion between hypotheses pertaining to training, and those pertaining to sociodemographical variables. At the end of the reading, I don’t see how the “collective intelligence” framework even relate to the present work. I don’t think the manuscript is suitable for publication in its present form.

Major points

1. The references in the Introduction section are outdated. A lot of work has been done in the last decades to understand the steps of ToM development. Please add more recent references, and more nuance in some assertions (for example page 6: “Due to the proposed reliance on instinct in the process, we believe ToM is best attributed to System 1 thinking, theorized by Kahneman (2011).).” See also major point number 5 on a related matter.

2. Page 6: I don’t see how referencing Kahneman’s framework is useful to understand the experiment or the results. Same for believers related data page 7. Same for mood related data page 7.

3. Page 11: “Analogous to the concept of diminishing returns”. Please provide a definition and references to back up the concept of “diminishing returns”.

4. Page 12: “Because mental state interpretation is quick, automatic, and universal”. Whether mental state understanding is automatic, universal or quick is the subject of entire research programs. I don’t think these questions are elucidated to this day. Please provide references to back up this assertion, and develop a theoretical argument to justify it, or either drop or adjust the assertion.

5. Page 14: “while more detail can be found in Authors (2018).” Please refrain from citing unpublished data. I couldn’t find the referenced work on the internet.

6. Page 19: “One-way ANOVAs were used to inspect the simple main effects of training condition,”… I don’t see the point of this analysis. The previous ANOVA on change scores, with Training Type as factor, already showed a main effect of High Vs Low Scorers. Why then use three different ANOVAs for each type of training to just show again this effect for each training? If the authors want to explore the marginal effect, they should instead run two additional ANOVAs: one for the High Scorers with Training Type as factor, and the same for the Low Scorers.

7. Page 19: “As predicted, the effect of previous experience with the RMET on Pre-RMET score was significant, t(426) = -2.41, p = .008. (Figure 4a). »… I don’t see how this was predicted in any of the hypotheses listed at the end of the Introduction. Please add the corresponding hypothesis in the Introduction and justify it, or correct this section. What was predicted is less progression between pre and post training for participants with previous exposure to the RMET, not lower scores at pre-training (see H3a). Please also edit the Discussion section, page 24. “In support of H3A, participants with previous experience taking the RMET performed significantly worse than participants who had no previous experience with the RMET.”

8. The authors say in the Introduction section that empathy could be a mediating factor explaining why women (or hypothetically marginalized groups), could perform better on the RMET. Why not include a measure of empathy in the present experiment? Could the authors discuss this matter?

9. Conclusion section. This section usually contains a single paragraph outlining the main results of the study. Please summarize.

10. The “Related work” section seems not appropriate for a peer reviewed article. It presents data that are not relevant to the experiment. Please extract the relevant information from this section, explicitly link them to your experiment and hypotheses, and drop irrelevant literature.

11. Since the present design is across subjects, it is possible that the Qualtrics randomization led to potential biases. For example, it is possible that a majority of female participants was in the “Repetitive Training” condition, then leading to the effect of training being confounded with the gender factor. Could the authors provide the repartition of participants and their characteristics in each training category? Could they also discuss this potential bias in the Discussion section?

Minor points

1. There is a typo in Baron-Cohen’s citations. There should be no space after “Baron-“. Please correct.

2. I don’t see the use of Table 1… Rather, a Table showing the characteristics of participants in each training group would be more useful.

3. Please move the Limitations and Assumptions section to the end of the paper.

4. Page 19: “Figure 2Error! Reference source not found.” Please correct.

5. Page 23: “, but a marginally significant, weak, and negative association was uncovered between overall RMET Score and overall average time per question”. I think that this part of the sentence refers to high skill participants, but this needs to be explicit. Please correct.

Reviewer #2: 1. Yes

2. Yes

3. Yes

4. Several parts of the paper should be rewritten.

The paper deals with an interesting topic. However, there are important issues that prevent from considering the paper ready for publication, at least in its actual form. The paper would benefit from a change of the structure. The introduction should be focused on the topic of the study, that is the training, and not on other irrelevant information such as benefits of higher Reading the Mind in the Eyes Ability or studies on theory of mind. I would suggest discussing in a critical way the training interventions reported in the literature. This could help the Authors explain the relevance and novelty of their study, at the moment it is not clear and defined.

The method section needs to be more specific. Several information such the length of the training should be added. The procedure section is very important, and I do not think is well presented. This makes difficult the interpretation of the results.

I do not think that the label “strategic training” is correct, I would suggest using something different that highlights the content of the intervention.

The limitations of the study should be moved to the end of the discussion section and the aims should be presented in the introduction. I would suggest reducing the number of hypotheses.

6. PLOS authors have the option to publish the peer review history of their article (what does this mean?). If published, this will include your full peer review and any attached files.

Reviewer #1: **Yes: **Matias Baltazar

Reviewer #2: No

---

## [Author Response · Author response to Decision Letter 0]

16 Dec 2021

We would like to thank the reviewers for their feedback and helpful comments to improve this work. They were extremely helpful and well targeted towards improving the paper. We assigned numbers to the comments so we could refer any of them back when there is similar comments from different reviewers. 

Editor

[E.1] Please include your full ethics statement in the ‘Methods’ section of your manuscript file. In your statement, please include the full name of the IRB or ethics committee who approved or waived your study, as well as whether or not you obtained informed written or verbal consent. If consent was waived for your study, please include this information in your statement as well.

[Author Response] We added the following statement: "This study was approved as exempt by the Institutional Review Board of Iowa State University (#18-075). Electronic informed consent was obtained from all participants.”

[E.2] Please ensure that you refer to Figure 1 and 3 in your text as, if accepted, production will need this reference to link the reader to the figure

[Author Response] The citation of Figure 1 has been added to the text. Figure 3 was deleted to address later review comments.

[E.3] Please ensure that your manuscript meets PLOS ONE's style requirements, including those for file naming. The PLOS ONE style templates can be found at https://journals.plos.org/plosone/s/file?id=wjVg/PLOSOne_formatting_sample_main_body.pdf and https://journals.plos.org/plosone/s/file?id=ba62/PLOSOne_formatting_sample_title_authors_affiliations.pdf

[Author Response] All style requirements have been implemented in the manuscript.

[E.4] Should your manuscript be accepted for publication, we will hold it until you provide the relevant accession numbers or DOIs necessary to access your data. If you wish to make changes to your Data Availability statement, please describe these changes in your cover letter and we will update your Data Availability statement to reflect the information you provide.

[Author Response] We have an Open Science Framework repository for this study’s data linked here: https://osf.io/bd4rp/

Reviewer #1

 [R1.1] The references in the Introduction section are outdated. A lot of work has been done in the last decades to understand the steps of ToM development. Please add more recent references, and more nuance in some assertions (for example page 6: “Due to the proposed reliance on instinct in the process, we believe ToM is best attributed to System 1 thinking, theorized by Kahneman (2011).).” See also major point number 5 on a related matter.

[Author Response] More recent references have been added to the introduction in the areas of training and the potential for training to be useful in creating better Human-AI collaboration. The assertion on page 6 has been removed as it was unnecessary when introducing ToM. Major point 5 is concerned with a potentially missing reference, Authors, 2018. This author names have been redacted for blind review, but the paper has been published.

[R1.2] Page 6: I don’t see how referencing Kahneman’s framework is useful to understand the experiment or the results. Same for believers related data page 7. Same for mood related data page 7.

[Author Response] Referencing Kahneman's framework is not necessary when introducing ToM, the experiment, or the results. Therefore, the assertion on page 6 has been removed. As part of the effort to the streamline the focus of the manuscript on the training, we have edited the manuscript to no longer include mentions of the research on the impact of belief in a god and mood, and other demographic variables, on the ability to engage in mindreading, or reading the mind in the eyes. 

[R1.3] 3. Page 11: “Analogous to the concept of diminishing returns”. Please provide a definition and references to back up the concept of “diminishing returns”.

[Author Response] The connection to "diminishing returns" was unnecessary in explaining the hypothesis. Therefore, the statement has been removed

[R1.4] 4. Page 12: “Because mental state interpretation is quick, automatic, and universal”. Whether mental state understanding is automatic, universal or quick is the subject of entire research programs. I don’t think these questions are elucidated to this day. Please provide references to back up this assertion, and develop a theoretical argument to justify it, or either drop or adjust the assertion.

[Author Response] The assertion has been removed from the paper as it was unnecessary in explaining the inclusion of H4.

[R1.5] 5. Page 14: “while more detail can be found in Authors (2018).” Please refrain from citing unpublished data. I couldn’t find the referenced work on the internet.

[Author Response] This is a published paper, and was referenced as "Authors" for the blind review. It was published in the proceedings of HFES 2018

[R1.6] Page 19: “One-way ANOVAs were used to inspect the simple main effects of training condition,”… I don’t see the point of this analysis. The previous ANOVA on change scores, with Training Type as factor, already showed a main effect of High Vs Low Scorers. Why then use three different ANOVAs for each type of training to just show again this effect for each training? If the authors want to explore the marginal effect, they should instead run two additional ANOVAs: one for the High Scorers with Training Type as factor, and the same for the Low Scorers.

[Author Response] Reviewer 1's suggestion to reduce the complexity of the results by running two separate ANOVAs to replace the three one-way ANOVAs is a clear improvement, and as such, we have redone our analysis to explore the marginal effects of score on training type and updated the discussion and results as necessary.

[R1.7] Page 19: “As predicted, the effect of previous experience with the RMET on Pre-RMET score was significant, t(426) = -2.41, p = .008. (Figure 4a). »… I don’t see how this was predicted in any of the hypotheses listed at the end of the Introduction. Please add the corresponding hypothesis in the Introduction and justify it, or correct this section. What was predicted is less progression between pre and post training for participants with previous exposure to the RMET, not lower scores at pre-training (see H3a). Please also edit the Discussion section, page 24. “In support of H3A, participants with previous experience taking the RMET performed significantly worse than participants who had no previous experience with the RMET.”

[Author Response] Reviewer 1 is correct that the results discussed on page 19 and visualized in Figure 4a is not related to H3A. This text was included mistakenly, and was meant to, instead, give more context to the findings (similar to the distribution of genders across conditions) and the (new) placement of the other demographic variable discussion. To correct this, as well as address Reviewer 2's comment about paring down the arguments to those which are essential, all of figure 4 has been removed.

[R1.8] The authors say in the Introduction section that empathy could be a mediating factor explaining why women (or hypothetically marginalized groups), could perform better on the RMET. Why not include a measure of empathy in the present experiment? Could the authors discuss this matter?

[Author Response] Reviewer 2 suggested that we reduce the number of hypotheses and restructure the arguments of the introduction. As we did not measure Empathy, we have dropped hypothesis H4 and the related discussion of empathy as a mediating factor. 

[R1.9] Conclusion section. This section usually contains a single paragraph outlining the main results of the study. Please summarize.

[Author Response] We have rewritten the Conclusion section to a single paragraph that summarizes the results and suggests future work.

[R1.10] The “Related work” section seems not appropriate for a peer reviewed article. It presents data that are not relevant to the experiment. Please extract the relevant information from this section, explicitly link them to your experiment and hypotheses, and drop irrelevant literature.

[Author Response] The Related Work section has been re-structured to focus on training. Other topics, such as the effect of demographics on RMET score, have been removed. 

[R1.11] 11. Since the present design is across subjects, it is possible that the Qualtrics randomization led to potential biases. For example, it is possible that a majority of female participants was in the “Repetitive Training” condition, then leading to the effect of training being confounded with the gender factor. Could the authors provide the repartition of participants and their characteristics in each training category? Could they also discuss this potential bias in the Discussion section?

[Author Response] In the (newly named) Data Integrity section (under Experimental Methods), we have added a Table that contains a description of participant characteristics by condition (Table 1). In addition we have added a statement indicating that there is a relative balance of demographic groups across conditions.

[R1.12] 1. There is a typo in Baron-Cohen’s citations. There should be no space after “Baron-“. Please correct.

[Author Response] Thank you for catching this. It has been corrected.

[R1.13] . I don’t see the use of Table 1… Rather, a Table showing the characteristics of participants in each training group would be more useful.

[Author Response] Thank you for this suggestion. We have added a table (Table 1) that gives counts by gender, race/ethnicity, native English speaker status, US-native status, and age for each condition.

[R1.14] 3. Please move the Limitations and Assumptions section to the end of the paper.

[Author Response] The Limitations and Assumptions section has been moved to the end of the Discussion section, just before the (summary) Conclusion.

[R1.15] 4. Page 19: “Figure 2Error! Reference source not found.” Please correct.

[Author Response] The Figure 2 citation has been corrected

[R1.16] 5. Page 23: “, but a marginally significant, weak, and negative association was uncovered between overall RMET Score and overall average time per question”. I think that this part of the sentence refers to high skill participants, but this needs to be explicit. Please correct.

[Author Response] That is correct, the second part of the sentence refers to high skill participants. The sentence has been rewritten to read, "but a marginally significant, weak, and negative association was uncovered between overall RMET Score and overall average time per question (τb = -.11, p = .09) for high-skill participants."

Reviewer #2

 [R2.1] The paper would benefit from a change of the structure. The introduction should be focused on the topic of the study, that is the training, and not on other irrelevant information such as benefits of higher Reading the Mind in the Eyes Ability or studies on theory of mind. I would suggest discussing in a critical way the training interventions reported in the literature. This could help the Authors explain the relevance and novelty of their study, at the moment it is not clear and defined.

[Author Response] We have rewritten the Introduction to focus on the central theme of the paper: can RMET be trained. We have added more discussion on previous work in training. The section on the benefits of higher RMET score has been removed from the literature review, as well as the material on demographics.

[R2.2] The method section needs to be more specific. Several information such the length of the training should be added.

[Author Response] We have added detail to the method section including: ethics statement, the length of the experiment overall, the length of each condition, limitations, and participant demographics by condition. 

[R2.3] The procedure section is very important, and I do not think is well presented. This makes difficult the interpretation of the results.

[Author Response] The procedure section has been restructured to better convey at what point in the study participants were split into conditions and the differences among the three conditions.

[R2.4] I do not think that the label “strategic training” is correct, I would suggest using something different that highlights the content of the intervention.

[Author Response] We've changed the condition label to "naturalistic" to reference the trainings' focus on identifying emotions in naturalistic settings (i.e., observing human faces or anthropomorphic faces in more context, including the addition of a mouth or narrative arc).

[R2.5] The limitations of the study should be moved to the end of the discussion section 

[Author Response] The limitations were moved to the end of the Discussion section.

[R2.6] the aims should be presented in the introduction

[Author Response] The aims of the study have been added to the last paragraph of the Introduction, and reads, "The aim of this work was to develop and test methods of training individuals to perform better on the RMET."

[R2.7] I would suggest reducing the number of hypotheses.

[Author Response] Upon reviewing our hypotheses, we have removed H4. This information deserves its own fully-fledged study, and it is beyond the scope of our original question about training. Thus, because of the breadth of our study, the demographic variables are no longer included in the hypotheses, only as discussion points. We also have removed H3A, as this hypothesis distracted from the focus of the paper.

---

## [Decision Letter · Decision Letter 1]

3 Feb 2022

PONE-D-21-26174R1An Evaluation to Determine if Reading the Mind in the Eyes Scores Can Be Improved Through TrainingPLOS ONE

Dear Dr. Stonewall,

Thank you for submitting your manuscript to PLOS ONE. After careful consideration, we feel that it has merit but does not fully meet PLOS ONE’s publication criteria as it currently stands. Therefore, we invite you to submit a revised version of the manuscript that addresses the points raised during the review process.

Thank you for carefully revising the manuscript based on the feedback received. One reviewer has raised additional or remaining concerns, which we feel should be addressed. Please see their suggestions below for enhancing the reproducibility and clarity of the study.

We look forward to receiving your revised manuscript.

Kind regards,

Hanna Landenmark

Senior Editor

PLOS ONE

Journal Requirements:

Reviewers' comments:

Reviewer's Responses to Questions

**Comments to the Author**

1. If the authors have adequately addressed your comments raised in a previous round of review and you feel that this manuscript is now acceptable for publication, you may indicate that here to bypass the “Comments to the Author” section, enter your conflict of interest statement in the “Confidential to Editor” section, and submit your "Accept" recommendation.

Reviewer #1: (No Response)

Reviewer #2: All comments have been addressed

2. Is the manuscript technically sound, and do the data support the conclusions?

Reviewer #1: Partly

Reviewer #2: Yes

3. Has the statistical analysis been performed appropriately and rigorously? 

Reviewer #1: Yes

Reviewer #2: Yes

4. Have the authors made all data underlying the findings in their manuscript fully available?

Reviewer #1: No

Reviewer #2: Yes

5. Is the manuscript presented in an intelligible fashion and written in standard English?

Reviewer #1: No

Reviewer #2: Yes

6. Review Comments to the Author

Reviewer #1: General comment

The authors really put effort into editing their manuscript. However, I think some points need clarification. Also, the structure of the paper is still unorthodox, which is a little confusing for readers.

Major points

1. The authors have added sentences pertaining to human-human interaction or human to AI interaction. What is the rationale for this? Please provide a rationale, if possible with references, or drop this argument.

2. I am still not satisfied with the “Related Work Section”. It really hinders the interest of your work as it looks more like a patchwork of data rather than a well-constructed introduction. I never found such section in any published article I ever read. I think it would really be better if the authors just picked the relevant parts of this section to inject them in a more streamlined Introduction Section.

3. I don’t understand the rationale for H4. If the authors really follow the line of reasoning in the experiments by Tracy and Robins (2008), they should hypothesize just the opposite, or maybe that there is no correlation between RT and accuracy... I find the H4 hypothesis very confusing and counterintuitive.

4. Page 18: “the RMET has demonstrated a strong ability to identify individuals with impaired social intelligence but otherwise normal cognitive intelligence”. This phrasing could be interpreted as offensive for people with Autism Spectrum Disorder. Please use "typical" instead of "normal", or another similar word. Please use another expression than "impaired social intelligence". The neurodiversity movement is all about being considered as different people and not as impaired or incomplete beings. So please, if you mention this idea, think about the people involved and how they will interpret your phrasing.

5. Page 18: “There are a few known limitations of the RMET, etc.” and the two following paragraphs. These paragraphs are interesting but should be integrated in the Introduction section. Usually, in the Methods section, when the literature guides choices regarding the methods, references are briefly provided with a one sentence rationale. If the rationale requires a full paragraph, it should be fully developed in the Introduction, and then briefly reminded in the Methods, where the relevant procedure is described.

6. Page 19: “Lastly, The RMET involves the association of facial expressions to high-level emotion words which may be difficult for even native English speakers”. Please add at least one reference to back up what is said here.

7. Page 23: “One-way ANOVAs were used to inspect the simple marginal effects, etc.” I don't understand why the authors would want to explore marginal effects. Usually one would want to test for differences (at p<.05) and report marginal effects when they are found instead.

8. Page 25: “or it could be evidence of regression to the mean, as explained by Kahneman”. I don't understand this sentence. What is regression to the mean in this context?

Reviewer #2: (No Response)

7. PLOS authors have the option to publish the peer review history of their article (what does this mean?). If published, this will include your full peer review and any attached files.

Reviewer #1: **Yes: **Matias Baltazar

Reviewer #2: No

---

## [Author Response · Author response to Decision Letter 1]

18 Mar 2022

Author Responses to Address Reviewer Comments

Paper Title: An Evaluation to Determine if Reading the Mind in the Eyes Scores Can Be Improved Through Training 

Submitted to: PLOS One

Manuscript Number: PONE-D-21-26174

We would like to thank the reviewers for their feedback and helpful comments to improve this work. They were extremely helpful and well targeted towards improving the paper. We assigned numbers to the comments so we could refer any of them back when there is similar comments from different reviewers. 

Editor

[E.1] Please review your reference list to ensure that it is complete and correct. If you have cited papers that have been retracted, please include the rationale for doing so in the manuscript text, or remove these references and replace them with relevant current references. Any changes to the reference list should be mentioned in the rebuttal letter that accompanies your revised manuscript. If you need to cite a retracted article, indicate the article’s retracted status in the References list and also include a citation and full reference for the retraction notice.

[Author Response] We have reviewed the reference list to ensure it is complete and correct. 

Reviewer #1

 [R1.1] The authors have added sentences pertaining to human-human interaction or human to AI interaction. What is the rationale for this? Please provide a rationale, if possible with references, or drop this argument.

[Author Response] In recent years, researchers have been looking to guide the creation of AI agents which better approximate human social intelligence. One of the threads of this research leads to Theory of Mind, which underlies the Reading the Mind in the Eyes Test. The sentences pertaining to human-human and human-AI interaction were added to demonstrate the utility of ToM training. The sentences pertaining to human-AI interaction have been rewritten, and another reference [10] added: "In human-AI interactions, the machine’s ability to attribute mental states to others greatly enhances the quality of the interaction [10]. As such, AI researchers have turned to ToM as a way to provide automations with the ability to process human facial expression data in real time [10]"

[R1.2] I am still not satisfied with the “Related Work Section”. It really hinders the interest of your work as it looks more like a patchwork of data rather than a well-constructed introduction. I never found such section in any published article I ever read. I think it would really be better if the authors just picked the relevant parts of this section to inject them in a more streamlined Introduction Section.

[Author Response] The separate "Related Work" section has been removed. The relevant content from this section has been integrated into the Introduction. Additionally, information that was redundant with the content of the introduction was removed. 

[R1.3] I don’t understand the rationale for H4. If the authors really follow the line of reasoning in the experiments by Tracy and Robins (2008), they should hypothesize just the opposite, or maybe that there is no correlation between RT and accuracy... I find the H4 hypothesis very confusing and counterintuitive.

[Author Response] Thank you for drawing this to our attention. We have altered the description of our reasoning to clarify that H4 is meant not to take Tracy and Robins line of reasoning and further it, but to attempt to duplicate or validate their findings under different contexts, to combat the ongoing "replication crisis" within social sciences.

[R1.4] Page 18: “the RMET has demonstrated a strong ability to identify individuals with impaired social intelligence but otherwise normal cognitive intelligence”. This phrasing could be interpreted as offensive for people with Autism Spectrum Disorder. Please use "typical" instead of "normal", or another similar word. Please use another expression than "impaired social intelligence". The neurodiversity movement is all about being considered as different people and not as impaired or incomplete beings. So please, if you mention this idea, think about the people involved and how they will interpret your phrasing.

[Author Response] We have replaced "normal" with "typical" and "impaired social intelligence" with "different social intelligence".

[R1.5] Page 18: “There are a few known limitations of the RMET, etc.” and the two following paragraphs. These paragraphs are interesting but should be integrated in the Introduction section. Usually, in the Methods section, when the literature guides choices regarding the methods, references are briefly provided with a one sentence rationale. If the rationale requires a full paragraph, it should be fully developed in the Introduction, and then briefly reminded in the Methods, where the relevant procedure is described

[Author Response] These paragraphs have been integrated into the Introduction section.

[R1.6] Page 19: “Lastly, The RMET involves the association of facial expressions to high-level emotion words which may be difficult for even native English speakers”. Please add at least one reference to back up what is said here.

[Author Response] This statement has been changed to reflect that taking the RMET in a language outside of one's native language may impact score. Additional references have been added: “Lastly, The RMET involves the association of facial expressions to emotion words which may be difficult for participants whose native language is not English. The difficulty of completing the RMET outside of one’s native language and the utility of offering the test in multiple languages is evidenced by the translation of the test into French [48] and Spanish among others [49].”

[R1.7] Page 23: “One-way ANOVAs were used to inspect the simple marginal effects, etc.” I don't understand why the authors would want to explore marginal effects. Usually one would want to test for differences (at p<.05) and report marginal effects when they are found instead.

[Author Response] This was a hold-over from our last iteration. We thank you for your insight. The statement has been changed to read “Two ANOVAs were used to inspect the marginal effects of training condition at each level of Initial RME ability, with significance evaluated at an alpha level of .025 to adjust for the use of multiple tests.”

[R1.7] “or it could be evidence of regression to the mean, as explained by Kahneman”. I don't understand this sentence. What is regression to the mean in this context?

[Author Response] We have added text to explain the statement within the context of our study: “This could indicate that the training is simply not effective for people who are already good at Reading the Mind in the Eyes, or it could be evidence of regression to the mean, a statistical phenomenon showing that high performers regress toward the mean or average while low performers will more typically see performance gains relative to their starting measurement [47]. Regression to the mean, in this context, speaks to the observation that average scores for the top 25% performers were on average one-and-a-half points lower (1.45) in the post-RMET, while the bottom 25% saw net gains of near two points (1.7), bringing each group closer to the average than where they started.”

---

## [Decision Letter · Decision Letter 2]

12 Apr 2022

An Evaluation to Determine if Reading the Mind in the Eyes Scores Can Be Improved Through Training

PONE-D-21-26174R2

Dear Dr. Stonewall,

We’re pleased to inform you that your manuscript has been judged scientifically suitable for publication and will be formally accepted for publication once it meets all outstanding technical requirements.

Kind regards,

Thomas Suslow, Ph.D.

Academic Editor

PLOS ONE

Additional Editor Comments (optional):

Reviewers' comments:

Reviewer's Responses to Questions

**Comments to the Author**

1. If the authors have adequately addressed your comments raised in a previous round of review and you feel that this manuscript is now acceptable for publication, you may indicate that here to bypass the “Comments to the Author” section, enter your conflict of interest statement in the “Confidential to Editor” section, and submit your "Accept" recommendation.

Reviewer #1: All comments have been addressed

Reviewer #2: All comments have been addressed

2. Is the manuscript technically sound, and do the data support the conclusions?

Reviewer #1: Yes

Reviewer #2: Yes

3. Has the statistical analysis been performed appropriately and rigorously? 

Reviewer #1: Yes

Reviewer #2: Yes

4. Have the authors made all data underlying the findings in their manuscript fully available?

Reviewer #1: Yes

Reviewer #2: Yes

5. Is the manuscript presented in an intelligible fashion and written in standard English?

Reviewer #1: Yes

Reviewer #2: Yes

6. Review Comments to the Author

Reviewer #1: I thank the authors for the hard work adressing my concerns. I am satisfied with the responses and/or text editions regarding all my major and minor concerns.

I present my apologies to the authors for my previous R1.7 point (regarding the exploration of a marginal effect with ANOVAs). I just wasn’t careful in my rereading of the manuscript and missed that your point was to explore a marginal effect shown in a previous analysis. I hope my useless point didn’t cost you too much time.

I have just one minor point left. But I don’t think it is important enough to prevent the manuscript to be published so I will let the authors and the editor decide whether it should be adressed or not. I don’t think it is necessay that I assess the revised manuscript myself.

1. Page 7, Hypothesis 4. I am sorry but I think relevant theory is still lacking in order for readers to understand this hypothesis. I am not a big fan of this theory but maybe cite Kahneman System 1 System 2 framework ? As you suggest, maybe « face reading » mechanisms involved in the RMET should conform to a System 1 / intuitive and fast cognitive style? And not to a System 2 / slow and reflecting cognitive style ? And that if it is the case we should expect associations between short reaction times and good accuracy ?

Reviewer #2: (No Response)

7. PLOS authors have the option to publish the peer review history of their article (what does this mean?). If published, this will include your full peer review and any attached files.

Reviewer #1: **Yes: **Matias Baltazar

Reviewer #2: **Yes: **Elena Cavallini

---

## [Editor Report · Acceptance letter]

19 Apr 2022

PONE-D-21-26174R2 

An evaluation to determine if reading the mind in the eyes scores can be improved through training 

Dear Dr. Stonewall:

I'm pleased to inform you that your manuscript has been deemed suitable for publication in PLOS ONE. Congratulations! Your manuscript is now with our production department. 

Kind regards, 

on behalf of

Professor Thomas Suslow 

Academic Editor

PLOS ONE